

# Reconstruction of warm season temperatures in central Europe during the past 60,000 years from lacustrine GDGTs

Paul D. Zander[1], Daniel Böhl[1], Frank Sirocko[2], Alexandra Auderset[1,3], Gerald Haug[1,4], Alfredo Martínez-García[1]

[1]Climate Geochemistry Department, Max Planck Institute for Chemistry, Mainz, 55128, Germany
[2]Institute of Geosciences, Johannes Gutenberg-University, Mainz, 55122, Germany
[3]School of Ocean and Earth Science, University of Southampton, Southampton SO14 3ZH, United Kingdom
[4]Department of Earth Sciences, ETH Zurich, Zurich, 8092, Switzerland

*Correspondence to*: Paul D. Zander (paul.zander@mpic.de)

## Abstract

Millennial-scale climate variations during the last glacial period, such as Dansgaard–Oeschger (D/O) cycles and Heinrich events, have been extensively studied using ice core and marine proxy records. However, there is a limited understanding of the magnitude of these temperature fluctuations in continental regions, and questions remain about the seasonal signal of these climate events. This study presents a 60,000-year long temperature reconstruction based on branched glycerol dialkyl glycerol tetraethers (brGDGTs) extracted from lake sediments from the Eifel volcanic field, Germany. brGDGTs are bacterial membrane lipids that are known to have strong relationship with temperature, making them suitable for temperature reconstructions. We test several temperature calibration models on modern samples taken from soils and multiple maar lakes. We find a bias associated with water depth and anoxic conditions that can be corrected for by accounting for a brGDGT isomer that is only produced in anoxic conditions. The corrected temperature reconstruction correlates with proxy and model temperature record spanning the same time period, validating the calibration model. However, millennial-scale variability is significantly dampened in the brGDGT record, and in contrast to other northern hemisphere climate records, during several Heinrich stadials, temperatures actually increase. We demonstrate that these apparent discrepancies can be explained by the unique seasonal response of the brGDGT paleothermometer to temperatures of months above freezing (TMAF). Our data support the view that warm season temperatures in Europe varied minimally during the last glacial period, and that abrupt millennial-scale events were defined by colder, longer winters. Our continuous high-resolution temperature reconstruction provides important information about the magnitude of seasonal climate variability during the last glacial period that can be used to test climate models and inform studies of paleoecological change.

## 1 Introduction

Millennial-scale climate variations during the last glacial period (Dansgaard–Oeschger (D/O) cycles and Heinrich events) are well documented in ice core and marine proxy records (Rasmussen et al., 2014; Davtian and Bard, 2023; Martrat et al., 2007;



Dansgaard et al., 1993). Abrupt warming is recorded within decades in ice cores at the onset of Greenland interstadials (GIs), followed by gradual cooling, forming a repeated saw-shaped pattern (Rasmussen et al., 2014; North Greenland Ice Core Project members, 2004). Coarse-grained layers of ice-rafted debris in the North Atlantic are markers of Heinrich events (Heinrich, 1988), which are associated with pronounced declines in sea surface temperatures and major changes in oceanographic

conditions, including weakened Atlantic Meridional Overturning Circulation (AMOC) (Martrat et al., 2007; Davtian and Bard, 2023; Bohm et al., 2015) and expanded sea ice in the North Atlantic (De Vernal et al., 2006). Millennial-scale climate events can be detected in terrestrial records from Europe during the last glacial period including lake sediments and paleoecological records (Duprat-Oualid et al., 2017; Guiot et al., 1993; Wohlfarth et al., 2008; Ampel et al., 2010; Fletcher et al., 2010), loess (Prud'homme et al., 2022; Újvári et al., 2017), and speleothems (Genty et al., 2003; Spötl and Mangini, 2002; Genty et al.,

2010). Yet, there is a lack of continuous quantitative estimates of temperature fluctuations across these events from continental regions, limiting our understanding of the magnitude of potential millennial scale climate variability on continents. The sparse proxy data leads to divergent interpretations of the magnitude of temperature change associated with these millennial-scale events, and in particular the role of changing seasonal conditions in driving observed changes in proxy records.

Glycerol dialkyl glycerol tetraethers (GDGTs) provide a unique opportunity to quantitatively estimate past temperatures over continental regions. Branched GDGTs (brGDGTs) are bacterial membrane lipids and are found in a wide range of environments including lakes, soils, loess, and marine environments (Hopmans et al., 2004; Schouten et al., 2013; De Jonge et al., 2014; Xiao et al., 2016; Raberg et al., 2022). The source organisms for these lipids are not well-constrained, however recent research has shown that certain strains of Acidobacteria produce brGDGTs in culture (Halamka et al., 2023; Chen et

al., 2022). Temperature has repeatedly been shown to drive brGDGT distributions, particularly the degree of methylation, and this relationship can be demonstrated in laboratory experiments (Martínez-Sosa et al., 2020; Martínez-Sosa and Tierney, 2019; Halamka et al., 2023), global datasets (Raberg et al., 2022, 2021; Martínez-Sosa et al., 2021; Naafs et al., 2017; Dearing Crampton-Flood et al., 2020), and through *in situ* monitoring (Zhao et al., 2021). brGDGTs preserved in lake sediments can provide continuous, high-resolution records of past temperatures, and research in recent years has focused on expanding global

calibration datasets for the lacustrine brGDGT paleothermometer. Initial studies interpreted brGDGTs in lacustrine sediments as derived from soil (Hopmans et al., 2004); however it is now established that the majority of lacustrine brGDGTs are produced within the aquatic environment (Bechtel et al., 2010; Tierney and Russell, 2009; Tierney et al., 2010; Van Bree et al., 2020; Weber et al., 2018, 2015; Zhao et al., 2021; Wang et al., 2023). Recent global calibrations have shown that brGDGTs are sensitive to the temperature of months above freezing (TMAF), likely because bacterial growth is substantially lower below

freezing temperatures, and ice cover on lakes disconnects lacustrine brGDGT producers from atmospheric conditions (Dearing Crampton-Flood et al., 2020; Martínez-Sosa et al., 2021; Raberg et al., 2021; Cao et al., 2020). The two most recent lacustrine brGDGT-temperature calibrations have errors of 2.1-2.9 °C across a diverse set of lakes spanning the globe (Raberg et al., 2021; Martínez-Sosa et al., 2021), proving this proxy can effectively be utilized for quantitative past temperature reconstructions.






Despite the ubiquitous relationship between temperature and brGDGT distributions, challenges remain for interpreting paleoclimate records from brGDGTs. Factors other than temperature are known to affect brGDGT distributions, which may lead to biases in temperature reconstructions. These non-climatic factors include oxygen/water depth (Weber et al., 2018, 2015; Van Bree et al., 2020; Halamka et al., 2023; Stefanescu et al., 2021), salinity (Wang et al., 2021), pH (De Jonge et al., 2014;

Parish et al., 2023; Martínez-Sosa and Tierney, 2019), and conductivity (Raberg et al., 2021). Furthermore, changes in the sources of brGDGTs can dramatically impact temperature reconstructions; brGDGTs produced in soils will produce a warm bias when input to a lacustrine calibration equation (Ramos-Román et al., 2022; Martin et al., 2020). Within lake variations in bacterial communities could also produce different temperature signals, with a cold bias from GDGTs produced at depths below the thermocline (Van Bree et al., 2020; Weber et al., 2018; Stefanescu et al., 2021; Sinninghe Damsté et al., 2022).


In this study, we measured brGDGTs in a composite lake sediment record from Eifel maar lakes (Sirocko et al., 2021) to reconstruct TMAF at multi-centennial resolution. Modern samples were utilized to test brGDGT-temperature calibrations, and we use an isomer that is only produced in anoxic settings to correct for biases that occur when a large portion of brGDGTs are produced in the hypolimnion of stratified lakes. The temperature record provides a unique insight into millennial scale climate

variability and changing seasonal temperature gradients in central Europe during the past 60,000 years.

## 2 Methods and materials

### 2.1 Study area and sample collection

The Eifel volcanic field of western Germany features more than 250 Quaternary age scoria cones (Schmincke, 2007), and is well known for its maar basins, seven of which are currently lakes, while another >60 have been filled with sediments (Sirocko,

2016). The modern climate of the region is a temperate oceanic climate (Cfb) with cool winters (mean Jan temperature = 0 °C), warm summers (mean July temperature = 16 °C), and precipitation distributed throughout the year (mean annual precipitation = 817 mm) (Deutscher Wetterdienst, Climate Data Center). The modern landscape is a mosaic of agricultural areas, urban land use, and patches of deciduous broad-leaved forests or mixed coniferous broad-leafed forests.

The Eifel Laminated Sediment Archive (ELSA) project has systematically drilled maar features in the Eifel (Sirocko, 2016). Five maar sites (Table 1; Figure 1) are relevant for the current study: Auel (dry) Maar, Gemündener Maar, Weinfelder Maar, Holzmaar, and Schalkenmehrener Maar. Sediment cores obtained from Auel Maar (AU3, AU4), Holzmaar (HM3, HM4), and Schalkenmehrener Maar (SMF1, SMF2) were used to build a continuous 60,000 year record (ELSA-20 stack), which has previously been investigated for changes in vegetation (Sirocko et al., 2016, 2022), lake productivity (Sirocko et al., 2021),

floods (Brunck et al., 2016), and dust (Fuhrmann et al., 2021). The chronology for the ELSA-20 stack is based on a combination of varve counts, marker layers, pollen stratigraphy, radiocarbon ages and tuning of high-resolution $C_{org}$ data with the NGRIP



$\delta^{18}$O record (Sirocko et al., 2021, 2022). For our downcore analysis of GDGTs, 317 samples were analyzed, with each sample typically spanning 10 cm. On average (median), each sample represents ~43 years, and the interval between each sample midpoint is 116 years (range 14-1380 yr); the temporal resolution is lower for the period 24,000 yr b2k (years before 2000 100 CE) to present. Samples from Auel Maar represent the period 58,340-14,300 yr b2k, Holzmaar represents 14,200-690 yr b2k, and Schalkenmehrener Maar represents 630-0 yr b2k.

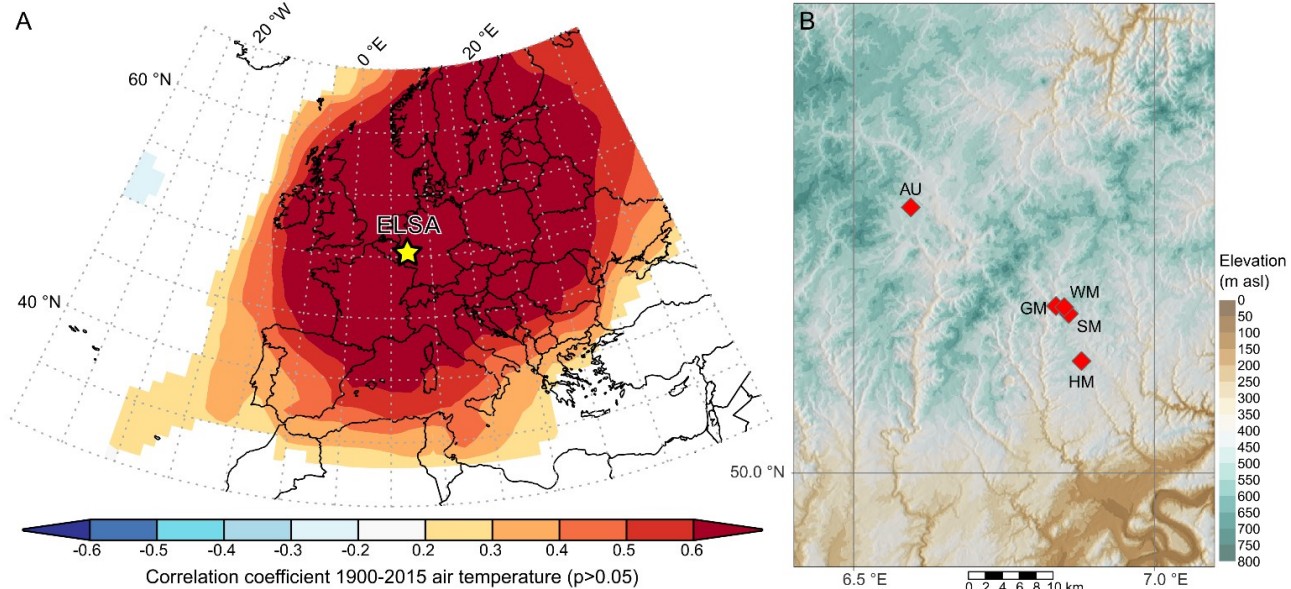

**Figure 1: Location map. A) Spatial correlation of annual temperature at ELSA site compared with regional surface temperatures over the period 1900-2015. Data from NOAA-CIRES-DOE 20th Century Reanalysis V3 (Compo et al., 2011; Slivinski et al., 2019).**
**B) Topographic map of sample sites for this study. AU= Auel Maar, GM = Gemündener Maar, WM = Weinfelder Maar, HM = Holzmaar, SM = Schalkenmehrener Maar. Topographic data from naturalearthdata.com.**

**Table 1: Summary of study sites and their properties. MAAT = Mean Annual Air Temperature and TMAF = Temperature of Months above Freezing. Temperature data refers to 1901-2016 period from CRU TS v4.01 dataset (Osborn and Jones, 2014).**

| Lake | ID | Longitude | Latitude | Altitude [m asl] | Water depth [m] | Diameter [m] | MAAT [°C] | TMAF [°C] | Soil | Lake surface | Lake core |
|---|---|---|---|---|---|---|---|---|---|---|---|
| Auel dry Maar | AU | 6.59502 | 50.2826 | 456 | - | 1325 | 8.01 | 8.77 | - | - | 285 |
| Holzmaar | HM | 6.87866 | 50.1191 | 425 | 21 | 272 | 8.22 | 8.97 | 4 | 4 | 39 |
| Schalkenmehrener Maar | SM | 6.85766 | 50.1694 | 420 | 21 | 528 | 8.25 | 9.00 | 4 | 18 | 15 |
| Gemündener Maar | GM | 6.83631 | 50.1777 | 407 | 38 | 309 | 8.34 | 9.09 | 3 | - | 2 |
| Weinfelder Maar | WM | 6.84997 | 50.1765 | 484 | 51 | 525 | 7.83 | 8.58 | 2 | - | - |

Modern samples of soils (n = 13) were taken from the vicinity of Gemündener Maar, Weinfelder Maar, Holzmaar, and Schalkenmehrener Maar. Lake surface sediments were also taken from Holzmaar, and Schalkenmehrener Maar; each surface sediment sample represents approximately the upper 4 cm of sediment. Four samples were taken from 19 m water depth in



Holzmaar, while a depth transect from 0.5-21.7 m water depth was obtained at Schalkenmehrener Maar (n = 18). Additionally, 13 samples from the upper 50 cm (representing sediments younger than 1900 CE) of sediment cores retrieved from
Gemündener Maar, Holzmaar, and Schalkenmehrener Maar were included in the modern set of samples (total of 48 samples) that were used to test GDGT calibrations against modern climate data.

**2.2 X-ray fluorescence core scanning**

Sediment cores from Holzmaar (HM4) and Auel Maar (AU4) were scanned with an Avaatech X-ray fluorescence (XRF) core scanner (Avaatech XRF Technology) at the Max Planck Institute for Chemistry in Mainz to obtain semi-quantitive elemental
data. The core face was smoothed and flattened using a blade, then covered with 4-μm-thin SPEX CertiPrep Ultralene film to avoid contamination and drying during measurements. XRF scans were performed using a rhodium anode tube with a voltage of 10 kV, a current of 550 μA, and counting time of 10 s. The measurement resolution was 0.5 mm. We utilize the counts of Ti as an indicator of terrigenous clastic sediment input, and the ratio of S/Ti as an indicator of hypolimnetic anoxia or sediment anoxia.

**2.3 GDGT extraction and measurement**

Samples were freeze-dried, homogenized, and then extracted using an accelerated solvent extractor (ASE) following the two-fraction method for lake sediments and soil of (Auderset et al., 2020). Between 0.2 and 3.51 g of dried sediment was weighed into an ASE cell with approximately 16 g of silica gel, which had previously been baked at 500 °C for 5 hours, then deactivated with 5% Milli-Q water and stored in $n$-hexane. A two-step extraction was performed with $n$-hexane and a 1:1
dichloromethane/methanol mix, which were flushed through the cells at 100 °C and 100 bar. The dichloromethane/methanol fraction contains the GDGT molecules. After extraction, a known amount (~700 ng) of internal $C_{46}$ GDGT standard (Patwardhan and Thompson, 1999) was added to the extracts. Solvents were then evaporated with a low-pressure centrifuge (Rocket by Genevac). Samples were re-dissolved in ~1 ml of a 98.5:1.5 mix of $n$-hexane/2-propanol and filtered with 0.4 μm PTFE filters. Subqequently, solvents were evaporated using a FlexiVap and samples re-dissolved in 300 μl of 9:1 $n$-hexane/2-
propanol, or a greater volume for particularly concentrated samples.

GDGTs were measured with an Agilent 1260 Infinity HPLC-MS (High Performance Liquid Chromatography Mass Spectrometer). The HPLC method was based on (Hopmans et al., 2016) and used a flow of 0.2 ml/min of 9:1 $n$-hexane/2-propanol through a silica column (UPLC BEH HILIC, 1.7μm) at 200 bar. Five μl were injected from each sample. A North
Atlantic Standard (Auderset et al., 2020) was run with each sample batch for quality control and to identify individual GDGT compounds. Data evaluation and peak integration was performed using Agilent MassHunter. The standard set of isoprenoid and brGDGT compounds were identified based on (Hopmans et al., 2016), and an additional isomer, IIIa'', which is specifically produced under low oxygen conditions, was also identified (Weber et al., 2018, 2015). The reproducibility of the method was assessed by repeated (n = 17) extraction and measurement of a single sediment sample (Table S1).



## 2.4 Temperature calibration and data analysis

Several calibration models exist for transforming brGDGT fractional abundances into temperature estimates, and the different calibrations can yield substantially different results. We tested several calibration models on our set of modern samples to assess their ability to produce temperatures that agree with instrumental temperatures. We used the CRU TS v4.01 dataset for modern temperature (representing the period 1901-2016; (Osborn and Jones, 2014)). We corrected for elevation differences at the five sample sites using a lapse rate of 6.6 °C/km calculated from nearby meteorological stations (German Weather Service). The altitude correction was also applied to reconstructed temperatures to ensure a homogenous reconstruction that represents the temperature at Auel Maar (465 m above sea level; asl). The following calibration models were assessed. We calculated mean annual air temperatures (MAAT) using the soil-based calibrations of De Jonge et al. (2014) using the MBT'$_{5Me}$ index and Index 1, respectively:

$$MBT'_{5Me} = \frac{Ia + Ib + Ic}{Ia + Ib + Ic + IIa + IIb + IIc + IIIa} \tag{1}$$

$$MBT'_{5Me}\ Soil\ MAAT = -8.57 + 31.45 * MBT'_{5ME} \tag{2}$$

$$Index\ 1 = log_{10} \frac{Ia + Ib + Ic + IIa' + IIIa'}{(Ic + IIa + IIc + IIIa + IIIa')} \tag{3}$$

$$Index\ 1\ Soil\ MAAT = 5.05 + 14.86 * Index\ 1 \tag{4}$$

We also calculated the Bayesian calibration models, BayMBT, for soils (Dearing Crampton-Flood et al., 2020) and lakes (Martínez-Sosa et al., 2021), which are based on the MBT'$_{5Me}$ index and calibrated for TMAF (Temperature of Months Above Freezing). We calculated temperatures using the FROG$_0$ (random **F**orest **R**egression for Pale**O**MAAT using br**G**DGTs) calibration model for soils (Véquaud et al., 2022). We calculated temperatures from the African lakes calibration models from Russell et al. (2018) using Index 1:

$$Index\ 1\ Lake\ MAAT = 12.22 + 18.79 * Index\ 1 \tag{5}$$

and stepwise forward selection multivariate regression:

$$SFS\ Lake\ MAAT = 23.81 - 31.02 * IIIa - 41.91 * IIb - 51.59 * IIb' - 24.70 * IIa + 68.80 * Ib \tag{6}$$

We calculated temperatures using the multivariate calibration model of Raberg et al. (2021), hereafter referred to as Raberg21:



$$Raberg21\ Lake\ TMAF\ =\ 92.9(\pm15.98) + 63.84(\pm15.58) * \text{fIb}_{\text{Meth}}^2 - 130.51(\pm30.73) * \text{fIb}_{\text{Meth}} - 28.77(\pm5.44) *$$

$$\text{fIIa}_{\text{Meth}}^2 - 72.28(\pm17.38) \cdot \text{fIIb}_{\text{Meth}}^2 - 5.88(\pm1.36) \cdot \text{fIIc}_{\text{Meth}}^2 + 20.89(\pm7.69) \cdot \text{fIIIa}_{\text{Meth}}^2 - 40.54(\pm5.89) \cdot \text{fIIIa}_{\text{Meth}} -$$

$$80.47(\pm19.19) \cdot \text{fIIIb}_{\text{Meth}} \tag{7},$$

where the compound fractions are calculated according to their methylation set; see Raberg et al. (2021) for details.

To assess how brGDGT distributions varied in different types of samples and different sites, a principal component analysis (PCA) was calculated on the fractional abundances of brGDGTs; the FAs were calculated as centered-log ratios and scaled prior to the PCA calculation. To assess changing sources of GDGTs we calculated several indices. The ΣIIIa/ΣIIa ratio was modified from Xiao et al. (2016) and Martin et al. (2019), and has been used as an indicator of aquatic production of brGDGTs:

$$\frac{\sum IIIa}{\sum IIa} = \frac{IIIa + IIIa' + IIIa''}{IIa + IIa''} \tag{8}$$

The branched and isoprenoid tetraether index (BIT) compares the proportion of branched GDGTs to the crenarchaeol, and has been used as an indicator of soil input (Hopmans et al., 2004, 2016):

$$BIT = \frac{Ia + IIa + IIa' + IIIa + IIIa'}{Ia + IIa + IIa' + IIIa + IIIa' + Crenarchaeol} \tag{9}$$

The %GDGT-0 index was calculated using the formula of (Peaple et al., 2022) and used as an indicator of anaerobic methane-oxidizing archaea or methanogenic Euryarchaeota (Sinninghe Damsté et al., 2012).

$$\%GDGT - 0 = \frac{GDGT - 0}{GDGT - 0 + Crenarchaeol} * 100 \tag{10}$$

Statistical analyses were performed in R v4.3 (R Core Team, 2022). Pearson's r was used to assess the strength of correlations between timeseries and was calculated using the package GeoChronr (Mckay et al., 2021). Timeseries data were mapped to a normal distribution; p-values were adjusted to take into autocorrelation using the effective n method (Dawdy and Matalas, 1964) and were adjusted for multiple hypothesis testing using the false discovery rate method of Benjamini and Hochberg (1995).



# 3. Results and discussion

## 3.1 GDGT distribution in modern samples and implications for temperature calibrations

The compositions of brGDGTs in modern samples shows clear differences between samples taken from soils and lake sediments (Figure 2, Figure 3). Soils have a greater proportion of tetra-methylated (Ia-c) GDGTs as well as IIa and IIa'. The

hexamethylated compounds IIIa, IIIa', IIIa'', IIIb, and IIIb' are less abundant in soils. The samples taken from the shallow lake shore (depths < 1 m) have a composition that falls between soil and lake sediment. The modern lake sediments tend to have generally similar distributions compared to each other, however, some small differences can be observed. SM has less tetra-methylated GDGTs and more IIIa'' indicating a dominance of aquatic production, and particularly high production within the anoxic hypolimnion. A strong relationship can be observed between fIIIa'' and depth in the transect from

Schalkenmehrener Maar (Figure 4), consistent with previous studies that have concluded this isomer is produced under low oxygen conditions (Weber et al., 2018, 2015).

**Figure 2: Distribution of brGDGT compounds at different sites (for abbreviations see Figure 1) and among different sample types. Boxes represent the interquartile range (IQR) of the data, with the median depicted by a horizontal line. Whiskers represent the minimum and maximum of the data, excluding outliers that are greater than 1.5 times the IQR beyond the IQR.**

**Figure 3: Principal component analysis biplots. Panels A and C show principal components 1 and 2. Panels B and D show principal components 2 and 3. Filled symbols in A and B represent the centroids of the groups (site or sample type). Fill in panels C and D is Raberg21_adj temperatures.**

The test of calibration models reveals significant differences in GDGT-estimated temperatures (Figure 5a). The soil models underestimate modern temperatures, which is to be expected when applied to lake sediments; however, three of the four soil calibrations also underestimate modern temperatures for the soil samples. The FROG₀ model produces temperature estimates with low errors, but also low variability. The two calibrations derived from African lakes (Russell et al., 2018) overestimate modern temperatures for all sample types, likely due to the fact that these models are calibrated for MAAT (mean annual air temperature) and are not well-suited for temperate lakes that are seasonally ice covered. The two recent global lake calibrations



(Raberg et al., 2021; Martínez-Sosa et al., 2021) tend to underestimate modern temperatures for most of the lake sediment samples. Across all calibration models a consistent relationship can be observed between fIIIa'' and the offset between the GDGT-based temperature estimate and the modern instrumental temperature, with high values of fIIIa'' corresponding to a cold bias (Figure 5a). High fIIIa'' is indicative of greater production of brGDGTs within the anoxic hypolimnetic waters of a
stratified lake, which are substantially cooler than mixed surface layers, particularly in summer (Figure 4).

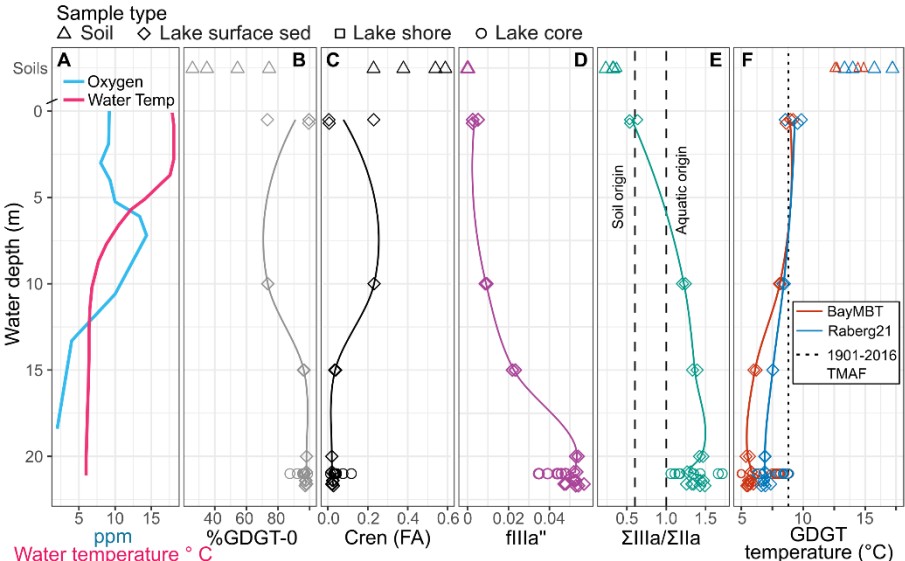

**Figure 4: Depth transect at Schalkenmehrener Maar. A) Water temperature and oxygen concentration measured in water column in May 2007 (Sirocko, 2012). B-F) GDGT data from surface sediment samples taken along a depth transect. B) %GDGT-0 index**
**showing relative abundance of methanogens and methanotrophs (Peaple et al., 2022; Sinninghe Damsté et al., 2012). C) Crenarchaeol fractional abundance of isoprenoid GDGTs indicating relative abundance of thaumarchaeota. D) Fractional abundance of IIIa'' relative to brGDGTs, indicating brGDGT production under anoxic conditions. E) ΣIIIa/ΣIIa ratio, a proxy for soil vs aquatic origin of brGDGTs. Thresholds for soil and aquatic origin from Xiao et al. (2016). F) brGDGT-inferred air temperatures (TMAF) from BayMBT (Martínez-Sosa et al., 2021) and Raberg21 (Raberg et al., 2021) calibrations.**

Based on the observed inverse relationship between fIIIa'' and the temperature offsets, we calculated regression models to estimate the temperature bias associated with hypolimnetic production of brGDGTs for each calibration model (Figure 5b) shows the linear regression models for the BayMBT lake model and the Raberg21 model. These regressions were found to be significant (p < 0.001), confirming our hypothesis that water-column stratification leads to a cold temperature bias in brGDGT temperature estimates. We utilize the regression models fit only to lake sediment samples because the soil samples have
temperature offsets that don't fit the linear trend and have distinct distributions from the lake samples. We then applied this correction to the temperature estimates from each calibration, which reduced the temperature offsets by accounting for the cold bias due to aquatic production of brGDGTs in bottom waters of the stratified maar lakes (Figure S1). Our data add to results from modern studies of brGDGTs at diverse lakes that have shown variable brGDGT composition along depth transects, suggesting that redox gradients could strongly influence brGDGT composition (Van Bree et al., 2020; Weber et al., 2018,




2015; Stefanescu et al., 2021). We propose that the fIIIa'' correction should be tested at other sites and is likely useful to improve the accuracy of brGDGT temperature calibrations, particularly at lakes that experience prolonged stratification with extensive low oxygen zones.

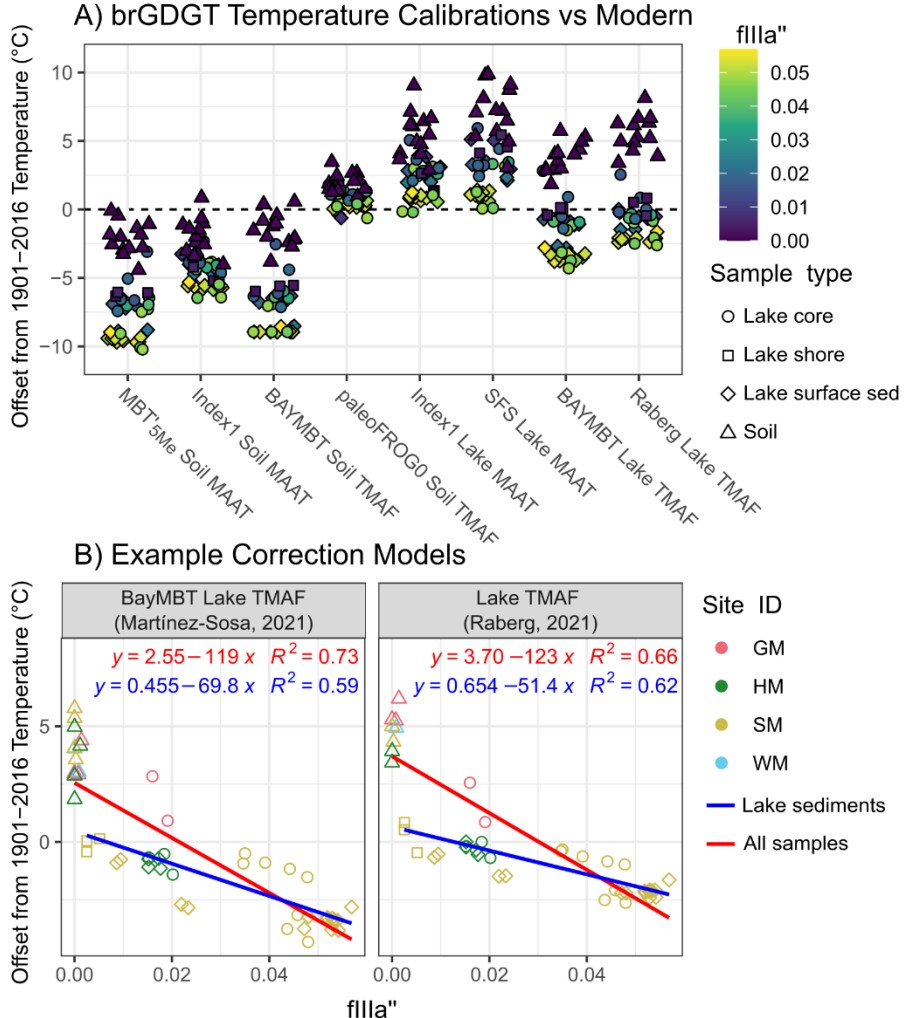

**Figure 5: A) Offsets between brGDGT-inferred temperatures and 1901-2016 temperature data for eight different calibration models.**
**Symbol shape corresponds to sample type, symbol fill corresponds to the fractional abundance of the IIIa'' isomer. B) Crossplots of temperature offsets and fIIIa'' for the two most recent lacustrine brGDGT temperature calibrations (Martínez-Sosa et al., 2021; Raberg et al., 2021). The regression model fit to lake samples (blue line) was used to correct the temperature reconstructions.**

**3.2 Assessing the impact of variable brGDGT sources on temperature reconstructions**

We reconstructed TMAF using both the BayMBT and Raberg21 model, with and without the adjustment for cold bias
associated with lake stratification and IIIa'' (Figure 6). In this section, we interrogate the different reconstructions to understand if changing sources of brGDGTs may have biased the reconstructions, in particular we assess evidence for variable





contributions of brGDGTs from soils, epilimnetic production (warmer surface waters), and hypolimnetic production (cooler bottom waters). All four reconstructions show broadly similar multimillennial scale patterns, with mild temperatures near or slightly cooler than present during early MIS 3, a colder period from 35-25 kyr b2k with temperatures around 2-3 °C cooler

than present, and very warm temperatures during the mid-Holocene (2-5 °C warmer than present). The magnitude of LGM to Holocene warming is comparable across calibration models. However, significant differences occur during millennial-scale climate perturbations. The BayMBT model shows several periods of pronounced cooling that are, counterintuitively, synchronous with interstadials recorded in Greenland ice cores (GIs 3, 4, 5, 6, 9, 10, 11 in particular). These interstadial periods are associated with proxy evidence for greater oxygen depletion in the hypolimnion (increased $C_{org}$, S/Ti, and fIIIa''; Figure

7); The adjusted BayMBT model shows a reduced magnitude of these cooling events, but nonetheless the contradictory cold interstadials remain. The Raberg21 model shows little change (cooling or warming) across these interstadial periods. This suggests that the BayMBT model is particularly sensitive to changes in brGDGT source, in particular the cold bias associated with increased production of brGDGTs in anoxic conditions. We interpret proxy evidence for increased primary productivity in the lake during interstadials to reflect a longer growing season. Furthermore, lake mixing likely also decreased either due to

thermal stratification or a reduction of winds, leading to an expanded anoxic zone, and leading to a cold bias in the MBT'$_{5Me}$ proxy.

We tested the correlation of each of the four reconstructions with prominent temperature records from the last glacial period: $\delta^{15}$N-based temperature estimates from Greenland ice cores (Kindler et al., 2014; Kobashi et al., 2017), alkenone-based sea

surface temperatures (SST) from the Iberian margin (Martrat et al., 2007), and $\delta$D-based temperature estimates from the EPICA ice core in Antarctica (Jouzel et al., 2007). Correlations were calculated with each timeseries averaged into bins of three different widths: 250, 500, and 1000 years (Table 2). The Raberg21 models correlate more strongly with each of the tested paleoclimate records compared to the BayMBT model. In particular, the BayMBT model shows weak correlations if the period is restricted to 15-59 kyr b2k. The Raberg21 training data includes 43 sites from high-latitude environments that are not

included in the BayMBT, and this may help improve the calibration for cold environments such as central Europe during the last glacial. The Raberg21_adj reconstruction shows higher correlations in all tests, and the correlations are significant (p_adj < 0.05) for all comparisons for the full period (0-59 kyr b2k). If the correlation test is restricted to the period prior to the deglaciation (15-59 kyr b2k), correlations decrease, but the Raberg21_adj model remains significantly correlated with the Greenland temperature data. The significant correlations indicate that the Raberg21_adj model is a valid temperature

reconstruction that captures the major climatic fluctuations of the past 60,000 years, and that the fIIIa'' correction method improves the accuracy of the reconstruction.



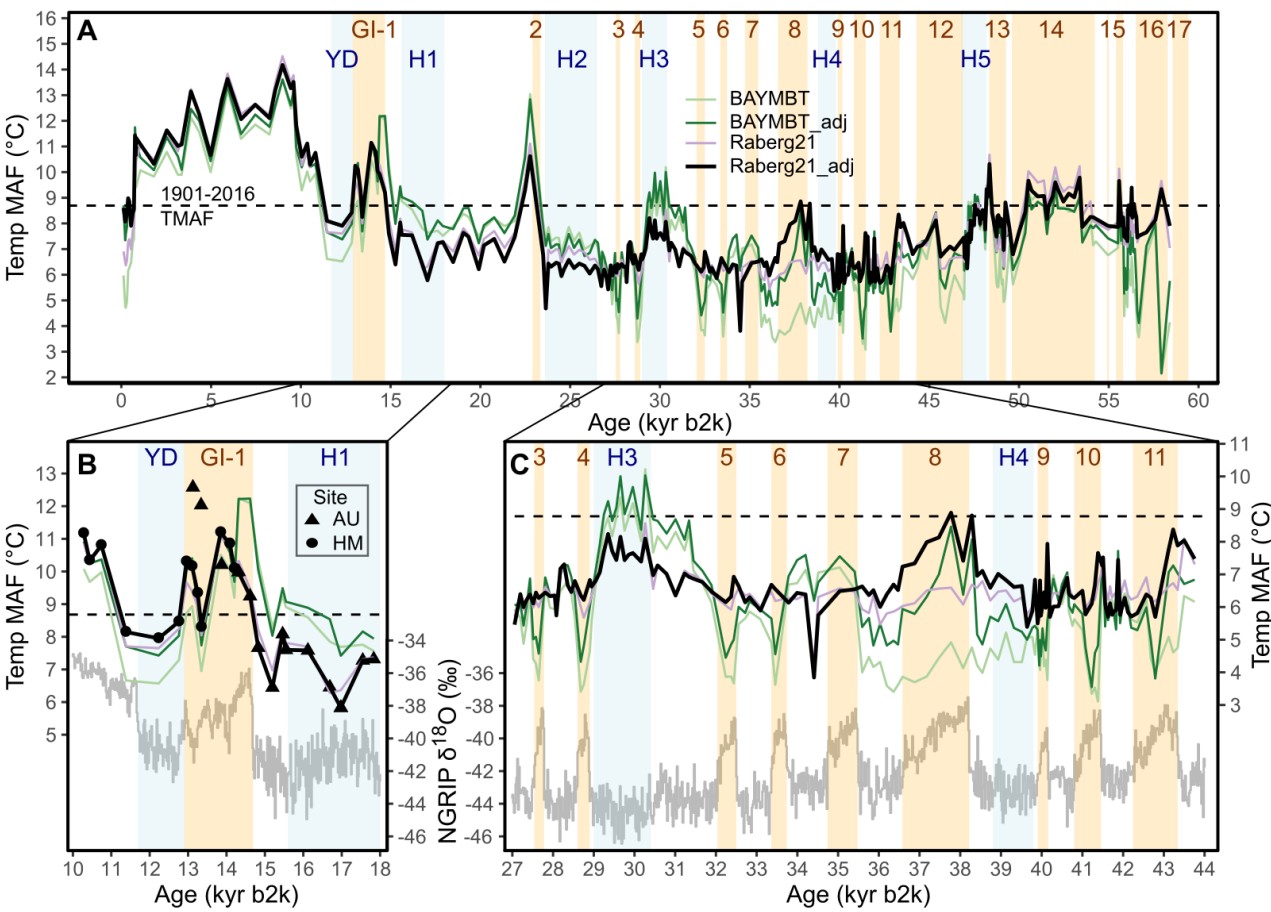

**Figure 6: Comparison of brGDGT temperature reconstructions with and without correction for anoxic production of brGDGTs.**
**Close-up panels show comparison with Dansgaard-Oeschger events in NGRIP ice core (gray, Rasmussen et al., 2014). Colored bars indicate Younger Dryas (YD), Greenland interstadials (GI) and Heinrich events (H).**



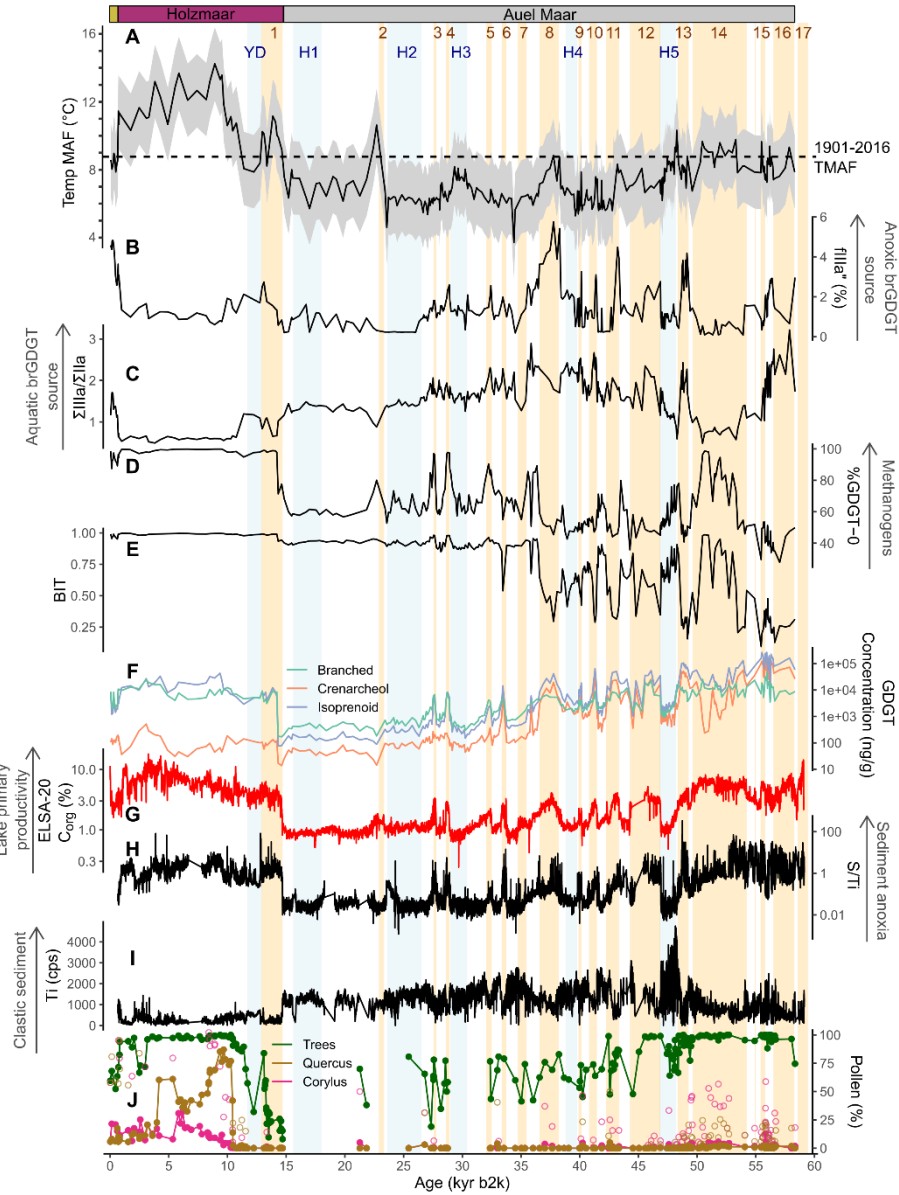

**Figure 7: Proxy data from ELSA sediments. A) brGDGT temperature reconstruction (Raberg21_adj), gray shading indicates calibration uncertainty from Raberg et al. (2021). B) Fractional abundance of IIIa'' isomer indicating anoxic production of brGDGTs. C) ΣIIIa/ΣIIa ratio, a proxy for soil vs aquatic origin of brGDGTs. D) %GDGT-0 index showing relative abundance of methanogens and methanotrophs. E) BIT index showing relative proportion of brGDGTs to Crenarcheol. F) Concentrations of brGDGTs, isoprenoid GDGTs and crenarchaeol (including crenarchaeol isomer). G) C$_{org}$ inferred from sediment reflectance (Sirocko et al., 2021). H) S/Ti ratio from XRF core scanning, used as a proxy for sediment/bottom-water anoxia. I) Ti counts, used as an indicator of lithogenic input. J) Tree pollen data with thermophilous species *Quercus* and *Corylus* highlighted (Sirocko et al., 2022). Non-filled circles in pollen plot represent 10x exaggerated pollen percentages.**



**Table 2: Correlation (Spearman's ρ) between brGDGT temperature reconstructions and paleoclimate timeseries. Bold values indicated p < 0.05. P-values were adjusted for autocorrelation of the timeseries using the effective-n method (Dawdy and Matalas, 1964) and for multiple hypothesis testing using the false discovery rate method (Benjamini and Hochberg, 1995). Greenland temperatures from Kindler et al. (2014) and Kobashi et al. (2017). North Atlantic SST from alkenone $U^{K'}_{37}$ on core MD01-2444 (Martrat et al., 2007). Antarctica temperatures from δD measured on EPICA core (Jouzel et al., 2007).**

|  | 0-59 kyr b2k | | | 15-59 kyr b2k | | |
|---|---|---|---|---|---|---|
|  | Greenland Temp | N. Atlantic SST | EPICA Temp | Greenland Temp | N. Atlantic SST | EPICA Temp |
| *250-year bins* | | | | | | |
| **Raberg21** | **0.63** | **0.66** | **0.59** | 0.27 | **0.35** | 0.32 |
| **Raberg21_adj** | **0.71** | **0.70** | **0.70** | **0.39** | **0.42** | **0.48** |
| **BayMBT** | **0.47** | 0.46 | 0.36 | 0.02 | 0.04 | -0.01 |
| **BayMBT_adj** | **0.54** | **0.51** | **0.46** | 0.10 | 0.06 | 0.06 |
| *500-year bins* | | | | | | |
| **Raberg21** | **0.64** | **0.68** | 0.59 | 0.29 | 0.38 | 0.25 |
| **Raberg21_adj** | **0.71** | **0.72** | **0.71** | **0.43** | 0.42 | 0.43 |
| **BayMBT** | **0.47** | **0.48** | 0.36 | 0.02 | 0.04 | -0.08 |
| **BayMBT_adj** | **0.54** | **0.52** | 0.45 | 0.08 | 0.02 | -0.01 |
| *1000-year bins* | | | | | | |
| **Raberg21** | **0.72** | **0.68** | 0.60 | 0.42 | 0.34 | 0.24 |
| **Raberg21_adj** | **0.77** | **0.72** | **0.71** | **0.53** | 0.44 | 0.44 |
| **BayMBT** | 0.50 | 0.46 | 0.35 | 0.07 | -0.01 | -0.12 |
| **BayMBT_adj** | **0.59** | **0.54** | 0.45 | 0.12 | 0.01 | -0.04 |

To further investigate what factors might influence mismatches between the GDGT temperature reconstructions and the Greenland temperature record, we calculated the offset between our four reconstructions and the Greenland temperatures after scaling the Greenland temperatures to match the mean and variance of the GDGT temperature estimates. These offsets were then correlated with paleoclimate and local proxy data (Figure 8). This analysis shows that the offsets with Greenland temperatures tend to be negatively correlated with the paleoclimate records, indicating that our reconstructions tend to have a cold bias during warm periods (and/or vice versa). However, this relationship is substantially weaker for the Raberg21 models compared to the BayMBT model. The BayMBT models show strong negative correlations with $C_{org}$ and fIIIa'', both of which are associated with greater aquatic production and hypolimnetic anoxia. The Raberg21 models also show negative correlations to these proxies, but the correlations are weaker, indicating the model is less sensitive to these non-climatic factors. The Raberg21_adj model is the only reconstruction that does not show a significant negative correlation between the offsets and fIIIa'', indicating that the cold bias associated with stratification has been mitigated due to the correction applied.





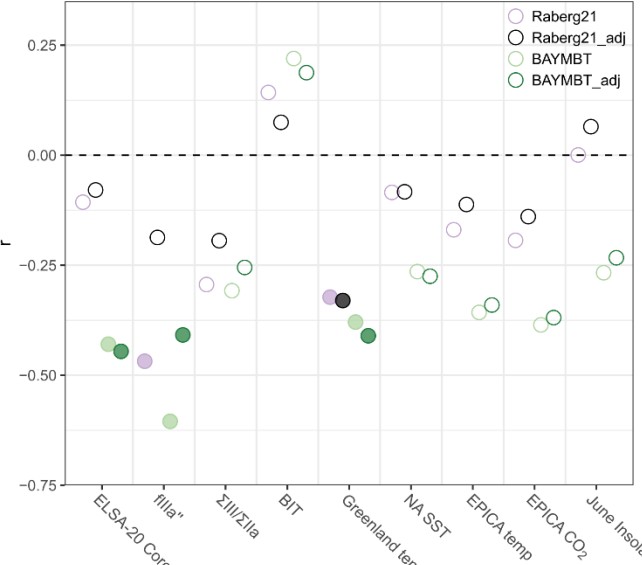

**Figure 8: Assessing which factors contribute to offsets between GDGT temperature reconstructions and the Greenland $\delta^{15}N$**
**temperature record after scaling mean and variance to match the GDGT temperatures (Kobashi et al., 2017; Kindler et al., 2014).**
**Plotted is the Pearson's r for these offsets and the proxy timeseries on the x-axis. Timeseries were placed in 250-year bins before the**
**correlation was calculated. Filled circles indicate significant correlations (p < 0.05 after adjusting for autocorrelation and multiple**
**hypothesis testing).**

Variations in aquatic vs soil sources of brGDGTs might influence the temperature reconstruction, as has been reported at other

European lakes (Martin et al., 2019, 2020; Ramos-Román et al., 2022). Correlations between the Greenland temperature offsets

and $\Sigma IIIa/\Sigma IIa$ and BIT could suggest that more (less) soil input is related to positive (negative) offsets. However, the

correlations are not statistically significant, indicating that changing contribution of terrestrial vs aquatic sourced GDGTs is

not a significant influence on the mismatch between the GDGT record and the Greenland record. Furthermore, although these

two indices have been used in previous studies as indicators of soil derived GDGTs, global compilations provide mixed

evidence of their utility. High BIT has been used as an indicator of terrestrial GDGT sources (Hopmans et al., 2004), but in a

global compilation of GDGT data (Martínez-Sosa et al., 2023), lake sediments actually have a higher average BIT than soils

(p < 0.05). The interpretation of low $\Sigma IIIa/\Sigma IIa$ as indicative of soil derived brGDGTs was originally proposed for marine

settings (Xiao et al., 2016), but has also been applied to lacustrine GDGTs (Martin et al., 2020). However, 65% of lake sediment

samples in a global compilation (Raberg et al., 2022) have $\Sigma IIIa/\Sigma IIa < 1$, suggesting that calibration models for lacustrine

sediments already account for some degree of soil input. Additionally, the $\Sigma IIIa/\Sigma IIa$ ratio is strongly negatively correlated

with temperature in global compilations of lacustrine brGDGTs (Raberg et al., 2022), in particular at values < 1. An additional

point of evidence that soil sources do not dominate the GDGT distribution is that no lake samples are identified as soil samples

by a machine learning algorithm that uses GDGT distributions to predict depositional environments (Martínez-Sosa et al.,

350      2023).



In our 60,000-year reconstruction, low ΣIIIa/ΣIIa and high BIT mainly occur during the interval represented by Holzmaar sediments. The composition of GDGTs in HM is unique in that it has low concentrations of Crenarchaeol, specifically produced by ammonium oxidizing Thaumarchaeota, and high GDGT-0, produced by methanogens. These characteristics indicate that
isoprenoid GDGTs have an aquatic (anoxic) source in HM. C/N ratios in organic matter in HM sediments are typically 9-11, also indicating a dominantly aquatic source for organics in HM (Anhäuser et al., 2014; Lücke et al., 2003). HM sediments plot between lake and soil samples from the Eifel in reduced dimensional space (Figure 3), however the similarity with soil is mainly driven by the first principle component, which is highly correlated to temperature, and therefore it is unclear if the similar composition of HM and soils could be due to HM representing the warmest period in the record (Holocene Thermal
Maximum; Kaufman and Broadman, 2023). PCs 2 and 3 show distinct differences between HM and soil samples. When compared to a global compilation of GDGTs (Martínez-Sosa et al., 2023), HM plots outside the range of soil samples in a PCA (Figure S2). Overall, there is evidence that HM GDGTs differ from the other sites with some evidence for a greater portion of soil-derived brGDGTs; however, there is also evidence that the GDGTs are still predominantly aquatic. The distinct composition of HM GDGTs could be related to a steep oxycline and extensive euxinic conditions in HM, as indicated by the
presence of pigments attributed to anoxygenic phototrophic bacteria in Holocene sediments of HM (S. Birlo, G. Wienhues, personal communication). The same pigments were not detected in sediments from Auel Maar (P. Zander, unpublished). The euxinic conditions in HM likely restrict the abundance of Thaumarchaeota (Callieri et al., 2016; Sinninghe Damsté et al., 2022), driving low BIT values. brGDGT production in the anoxic zone was apparently relatively low (low fIIIa'') in Holocene HM, and this points toward a greater proportion of brGDGTs produced in the warm epilimnion of HM or in catchment soils,
which could be a justification for anomalously warm temperatures in the HM segment. Similarity of the HM Holocene samples and SM lake shore samples (<1 m water depth) strengthen this interpretation (Figure 3Figure 4). The reasonable agreement (considering significant age uncertainty in the Auel Maar data) of reconstructed temperature from samples taken from both Auel Maar and Holzmaar during the Bolling-Allerod warm period (GI-1) is evidence that compatible results can be obtained from both sites despite substantially different sediment properties in this interval (Figure 6).


Modelled TMAF from a high-resolution global circulation model (HadCM3; (Leonardi et al., 2023; Beyer et al., 2020) agrees with brGDGT-inferred temperatures (Figure 9), though the model is not capable of capturing millennial-scale variations. The root mean square deviation between proxy and model is 2.19 °C, nearly identical to the calibration error of 2.14 °C. The Holocene segment from Holzmaar is the only prolonged period of deviation between the model and proxy data, which may be
attributed to changing brGDGT sources as described above. However, it is likely that the model output also underestimates TMAF to some extent during the Holocene thermal maximum (HTM; 5-11 kyr BP) due to an underestimation of vegetation expansion and subsequent albedo feedbacks during the HTM (Thompson et al., 2022). Proxy data from Europe consistently show warmer than present summer temperatures during the HTM (Kaufman and Broadman, 2023; Martin et al., 2020; Renssen et al., 2009; Kaufman et al., 2020), which is not observed in the model output.





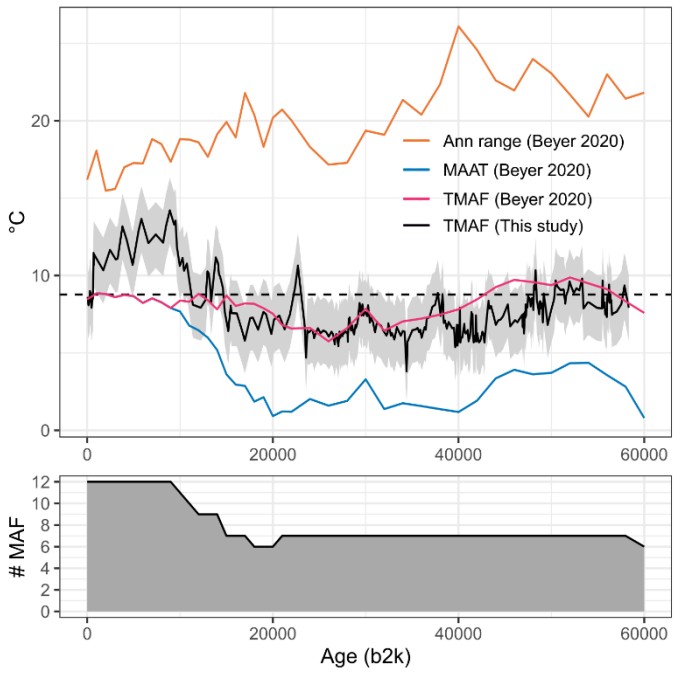


**Figure 9: Comparison of HadCM3 model simulations (Beyer et al., 2020; Leonardi et al., 2023) of seasonality, MAAT and TMAF with brGDGT temperature reconstruction.**

### 3.3 Comparison with regional paleoclimate records

The TMAF reconstruction shows multi-millennial patterns and magnitudes of change that are congruent with existing
temperature reconstructions from Europe (Figure 10). An early MIS-3 warm period occurs from 54-51 kyr b2k (GI-14) in
which TMAF is approximately the same as modern conditions (9 °C). This warm period corresponds with the prescence of
thermophilous tree species *Quercus* and *Corylus* in the study area (Figure 7). Evidence of summer temperatures near modern
levels during early MIS-3 can also be found from insect remains in N. Finland, the British Isles and France (Helmens, 2014)
and peak temperatures recorded in an Austrian speleothem (Spötl and Mangini, 2002). During Heinrich event 5 (H5), brGDGTs
record relatively warm temperatures in contrast to severely depressed SST in the North Atlantic at this time (Figure 10; Martrat
et al., 2007). Low ΣIIIa/ΣIIa, low $C_{org}$, and high Ti could indicate a shift towards more soil-derived brGDGTs during H5
(Figure 7), which would lead to a warm bias in the TMAF reconstruction. Alternatively, enhanced seasonality may have led
to higher TMAF despite cold long winters (Denton et al., 2022). Following H5, temperatures enter a cool phase from 44-38
kyr b2k, corresponding with a minimum of summer insolation, and including H4. GI-8 stands out as prolonged warm period
in our record, with ~2 °C of warming at the onset of the interstadial. After gradual cooling, TMAF averages 6.4 °C from 36-
30.5 kyr b2k. HE3 is again associated with an increase in TMAF, most likely associated with a prolonged freezing-season.



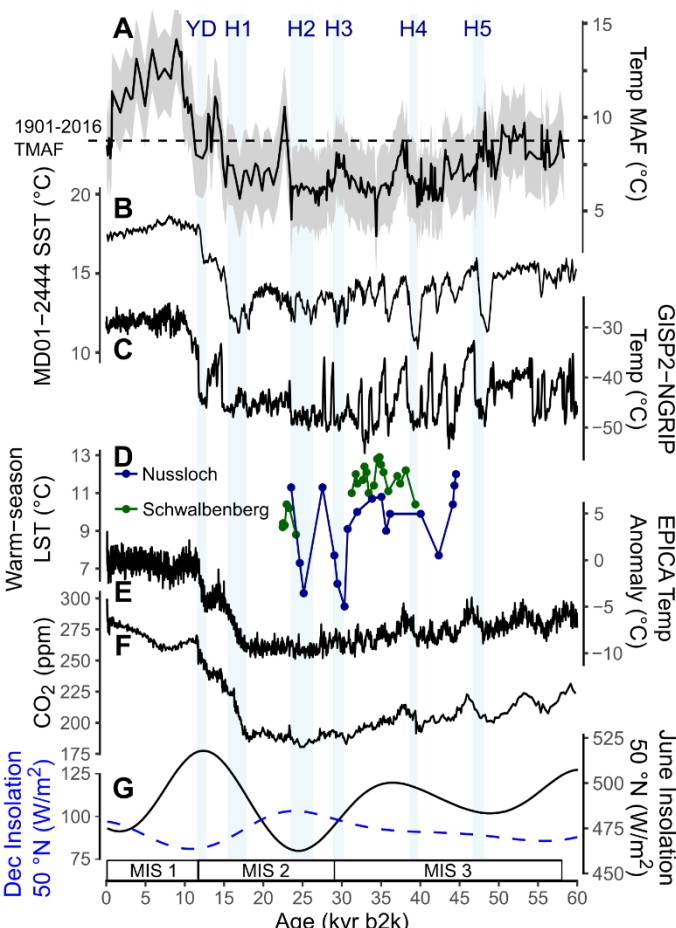

**Figure 10: Comparison of ELSA brGDGT temperature reconstruction with regional and global paleoclimate records. A) brGDGT temperature reconstruction (Raberg21_adj). B) North Atlantic SST from core MD01-2444 based on alkenone $U^{k'}_{37}$ proxy (Martrat et al., 2007). C) Greenland $\delta^{15}N$ temperatures (Kobashi et al., 2017; Kindler et al., 2014). D) Earthworm calcite grain warm season land surface temperatures (LST) from German loess (Prud'homme et al., 2022). E) Antarctica temperature anomalies from $\delta D$ (Jouzel et al., 2007). F) Atmospheric $CO_2$ estimated from the EPICA ice core (Bereiter et al., 2015). E) Incoming solar insolation for 50 °N during summer (black line) and winter (blue dashed line) (Laskar et al., 2004).**

From 29-24 kyr b2k TMAF reaches lowest levels of the record (~6.2 °C), in good agreement with the timing of maximal glacier extent in the Alps (Seguinot et al., 2018), British Isles and Baltic lowlands (Clark et al., 2009). The estimate of 2.6 °C cooling during the LGM compared to modern is substantially less than noble gas estimates of 9.1 °C cooling from the Paris Basin (Bekaert et al., 2023) and the results of a global paleoclimate data assimilation (~10 °C cooling at our site; Tierney et al., 2020). However, considering the stronger seasonal variability during the LGM, and thus longer freezing period, changes in TMAF are expected to be smaller than MAAT. Beetle (Coleoptera) remains from Auel Maar indicate that during the LGM, temperatures of the warmest month were 6 to 9.5 °C, whereas temperatures of the coldest month were -20 to -30 °C (Britzius and Sirocko, 2022), approximately double the annual temperature range compared to modern. Scaling the modern monthly climatology to match these temperatures would yield a TMAF between 5.8 and 4.3 ° C, which overlaps with our estimated



4.0-8.3 °C, taking into account calibration uncertainty (Figure 11). HadCM3 model output yields an TMAF of 6.3 °C over the 24-29 kyr period, nearly identical to the brGDGT estimate (Leonardi et al., 2023; Beyer et al., 2020). In the model output, the

number of months above freezing was 6-7 months during MIS 2-3 with an amplified seasonal temperature range of mean monthly temperature up to 26 °C (modern range is 16°C) (Figure 9). The enhanced seasonal range helps to explain the much smaller TMAF cooling compared to MAAT (Figure 11).

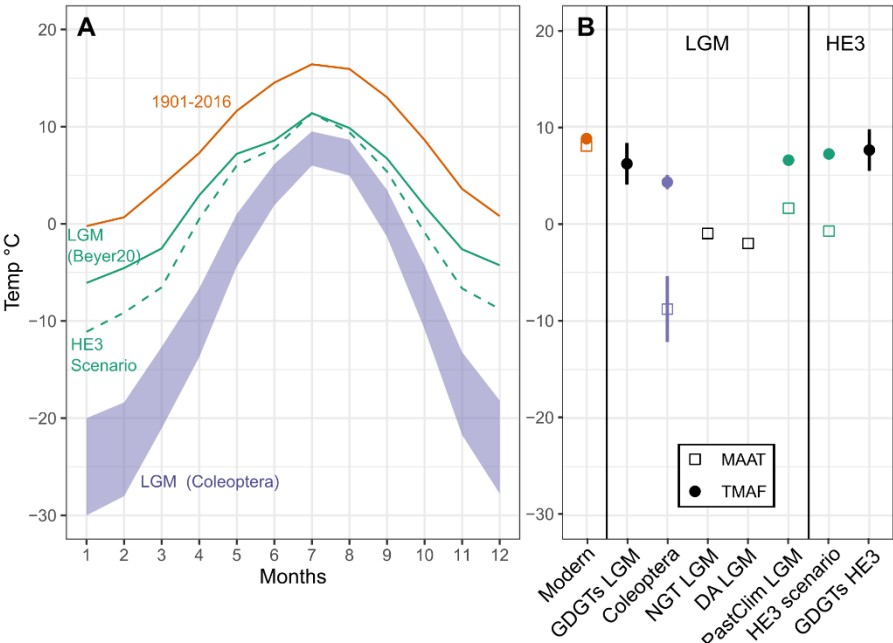

**Figure 11: A) Comparison of modern seasonal temperature range for the modern period, LGM based on climate models (Beyer et**
**al., 2020), LGM based on Coleoptera remains (Britzius and Sirocko, 2022), and a hypothetical Heinrich event scenario in which winter temperatures decrease while peak summer temperatures remain the same. B) Comparison of TMAF and MAAT for modern, LGM and HE3 based on different methods. NGT LGM = Noble Gas Temperatures from (Bekaert et al., 2023), DA LGM = data assimilation results from (Tierney et al., 2020). Notes: Coleoptera temperature range was obtained by scaling modern monthly temperatures to match the Coleoptera estimates of warmest and coldest months, the ranges represent likely values based on a simple**
**assumption about the monthly temperature pattern, but are not a calibrated estimate.**

Two data points suggest anomalously high TMAF near modern values during GI-2 (the ages of these samples are slightly younger than GI-2, but we attribute them to GI-2), which are suggestive of seasonally warm temperatures, though additional data would be needed to confidently confirm this interpretation. The pattern of deglacial warming captures major features such as the Bolling-Allerod warm period, Younger Dryas stadial, and rapid warming during the early Holocene. The Bølling–

Allerød (GI-1) TMAF is slightly warmer than modern and 2.8 °C warmer than H1. Chironomid-based summer temperature reconstructions from the alpine region and NW Europe suggest broadly similar magnitudes of warming from H1 to GI-1 (Heiri et al., 2015, 2014). Low sampling resolution precludes meaningful estimates of Younger Dryas cooling. Mid-Holocene temperatures average 12.3 °C, or 3.5 °C warmer than present, suggestive of a strong Holocene thermal maximum combined with strong seasonality in response to high summer insolation. The magnitude of the HTM is greater than estimated from



pollen-based summer temperature reconstructions (Mauri et al., 2015; Renssen et al., 2009; Martin et al., 2020), but is similar

to the magnitude of warming in chironomid-based reconstructions from the alpine region (Heiri et al., 2015).

### 3.4 Minimal millennial-scale variability in summer temperatures during the last glacial period

Our GDGT temperature reconstruction shows substantially dampened millennial-scale climate variability in comparison with
paleoclimate records from the North Atlantic region.

shows the temperature and warming estimated for each Greenland Interstadial (GI) and the early Holocene. For all but one GI
from 59-24 ky b2k (GIs 3-16), the warming is less than 1 ° C, which is less than the error of the temperature calibration (2.14
°C). We calculate a mean warming of 0.5 +/- 1.3 °C for GIs 1-16. These results suggest that Dansgaard-Oeschger (DO) cycles
had a minimal impact on warm-season temperatures in central Europe. This finding is surprising considering major changes
observed in aquatic productivity and redox conditions in Auel Maar and other lakes in western Europe (Wohlfarth et al., 2008)

during DO cycles. Additionally, pollen records indicate major vegetation shifts associated with DO cycles (Wohlfarth et al.,
2008; Moreno et al., 2014; Duprat-Oualid et al., 2017; Guiot et al., 1993), and shifts in $\delta^{18}$O (Spötl and Mangini, 2002; Moseley
et al., 2014) and $\delta^{13}$C (Genty et al., 2003) measured on speleothems indicate major changes to atmospheric circulation and
terrestrial ecosystems, respectively. However, our record supports the hypothesis that these changes were driven primarily by
variability in winter temperatures associated with expanded sea ice (Denton et al., 2005, 2022; Flückiger et al., 2008). Changes

in the duration of the growing season likely had a major impact on biotic proxies such as pollen and lake productivity.

Quantitative estimates of millennial-scale temperature fluctuations from terrestrial archives in Europe are rare beyond the past
15 kyr. Additionally, estimates of seasonal temperatures are challenging to interpret due to the difficulty of disentangling
multiple climate signals. Isotopic measurements of earthworm calcite granules from the Rhine Valley, Germany were used to

estimate approximately 1-4 °C of warming during the warmest months during GIs (Prud'homme et al., 2022). Clumped isotope
measurements on snails in Hungary indicate a difference of 4-7 °C between stadial and interstadial phases for the growing
season (Újvári et al., 2021). Temperatures estimated from diatom assemblages at Les Echets, France suggest 0.5-2 °C of
summer warming during GIs between 18-36 kyr BP (Ampel et al., 2010). Moraine records from Alpine glaciers provide
evidence of low millennial-scale temperature variability; simulations of Alpine ice extent fit the moraine records better when

forced by temperatures scaled to the Antarctica temperature record, rather than Greenland (Seguinot et al., 2018). Model
simulations of DO type variability also provide variable results. Two simulations done with LOVECLIM using perturbations
in Atlantic meridional overturning circulation (AMOC) yielded very different estimates for DO cycle temperature variability.
Using the LOVECLIM model, Van Meerbeeck et al. (2011), estimated 8 °C warming in Europe during the warmest month
and 17 °C warming during the coldest month for a stadial-interstadial transition during early MIS-3, whereas Menviel et al.

(2020) report 1.5 °C warming of MAAT over Europe associated with stadial-interstadial changes at 37 kyr BP, also using
LOVECLIM. A modeling study that isolated the impact of AMOC changes showed that, in Europe, spring temperatures
changed more than any other season in response to a shift from weak to strong AMOC, with 3.5 °C of warming in spring



compared 2 °C in summer (Flückiger et al., 2008). The variations in spring temperatures would have important implications
for the brGDGT proxy by changing the length of the non-freezing season.


**Table 3: Temperatures of warm events recorded by ELSA GDGT reconstruction and compared to temperature changes recorded in Greenland.**

| Time period | Start (yr b2k) | End (yr b2k) | GDGT TMAF (°C) | n (interstadial) | Warming from previous stadial (°C) | n (stadial) | Greenland Temp (°C) | Greenland warming (°C) |
|---|---|---|---|---|---|---|---|---|
| Early Holocene | 11700 | 8326 | 11.70 | 10 | 3.53 | 2 | -28.86 | 15.16 |
| GI-1 | 14692 | 12896 | 9.91 | 9 | 2.83 | 17 | -37.97 | 7.35 |
| GI-2 | 23340 | 22400 | 9.97 | 2 | 3.82 | 25 | -44.66 | 3.19 |
| GI-3 | 27780 | 27540 | 6.34 | 2 | -0.23 | 10 | -41.24 | 7.12 |
| GI-4 | 28900 | 28600 | 6.27 | 2 | -0.95 | 23 | -40.40 | 7.41 |
| GI-5 | 32500 | 32040 | 6.44 | 3 | 0.22 | 4 | -44.24 | 7.07 |
| GI-6 | 33740 | 33360 | 6.59 | 2 | 0.91 | 5 | -45.42 | 3.82 |
| GI-7 | 35480 | 34740 | 6.50 | 2 | 0.07 | 7 | -40.41 | 6.73 |
| GI-8 | 38220 | 36580 | 7.82 | 6 | 1.05 | 11 | -40.83 | 6.81 |
| GI-9 | 40160 | 39900 | 6.27 | 8 | 0.04 | 7 | -45.75 | 1.75 |
| GI-10 | 41460 | 40800 | 6.48 | 6 | 0.21 | 16 | -41.15 | 6.10 |
| GI-11 | 43340 | 42240 | 6.70 | 9 | -0.75 | 5 | -42.08 | 8.10 |
| GI-12 | 46860 | 44280 | 7.20 | 7 | -0.74 | 34 | -39.37 | 6.49 |
| GI-13 | 49280 | 48340 | 8.04 | 12 | 0.18 | 1 | -42.07 | 0.71 |
| GI-14 | 54220 | 49600 | 8.75 | 19 | 0.70 | 2 | -40.36 | 5.95 |
| GI-15 | 55800 | 55400 | 8.38 | 5 | 0.14 | 22 | -40.48 | 6.07 |
| GI-16 | 58280 | 56500 | 8.03 | 5 | 0.12 | 1 | -41.56 | 4.50 |

The GDGT temperature reconstruction shows a variable response to North Atlantic Heinrich Events. H5 and H3 are associated
with slight warmings in our reconstruction, and H1 has relatively mild temperatures. These results appear to be consistent with
major meltwater pulses recorded in sediments from the Bay of Biscay indicating melting of European ice sheets and glaciers
during Heinrich Events 1-3 (Toucanne et al., 2015), which would require relatively warm summers. Denton et al. (2022)
summarize existing evidence for relative warmth during Heinrich summers.

The seasonal response of lacustrine brGDGTs to the non-freezing season is a primary reason why our estimate of stadial-
interstadial temperature variability is lower than other records from Europe. Over the instrumental period, TMAF and MAAT
are nearly the same for our study area (8.77 °C vs 8.1 °C); however, during the last glacial period, with cooler temperatures
and increased continentality, the difference between MAAT and TMAF would have increased substantially (Figure 9). We
demonstrate that it is possible for the combination of cooler MAAT and increased seasonality to increase TMAF by simulating
a hypothetical HE3 monthly temperature pattern which has the July temperature simulated by HadCM3 for the LGM from



Beyer et al. (2020), but with a 5 °C greater seasonal range (Figure 11). Compared to the LGM values, this simulated Heinrich event would have MAAT 2.6 °C cooler, but TMAF is 1.1 °C warmer. This assumes a constant shape of the annual temperature cycle; however, particularly cold spring temperatures during Heinrich events (Flückiger et al., 2008) could enhance this effect. This response to the changing duration of the non-freezing period dampens variability in brGDGT temperature reconstructions

when colder temperatures are associated with increased seasonal amplitude. Temperature calibrations rely on an assumption that brGDGTs respond to TMAF, but detailed knowledge of seasonal production of brGDGTs might improve interpretation of temperature reconstructions, particularly because the organisms producing brGDGTs in lakes are not yet identified. Field monitoring studies of lacustrine brGDGT production suggests that production can be dominated by relatively short periods in the year and can be triggered by mixing events (Loomis et al., 2014; Van Bree et al., 2020), which could bias the temperature

signal to record conditions associated with lake mixing (typically 4-8 ° C for temperate lakes).

Overall, our data add to a growing body of evidence that changes in seasonality, and particularly winter climate, drove the abrupt millennial-scale climate events of the last glacial period, while summer temperatures remained relatively stable and moderate.

**4. Conclusions**

GDGT distributions measured on a composite sequence of lake sediments from the Eifel volcanic field, Germany provide a unique tool to reconstruct temperature fluctuations in central Europe during the past 60 kyr. A comparison of modern GDGT distributions in soils and lakes from the Eifel provides insights into the accuracy of the various published brGDGT-temperature calibrations. We find that lake stratification likely leads to a cold bias in brGDGT temperature estimates, and that this bias can

be corrected for using the relationship between the residuals of the temperature calibrations in this region and the IIIa'' isomer, which is exclusively produced in anoxic conditions. We find that the multivariate calibration model of Raberg et al., 2021 performs best, and after adjusting for the effect of lake stratification, produces a temperature reconstruction that resembles other temperature reconstructions from the past 60 kyr. The adjusted Raberg21 model is less affected by non-climatic factors, such as changing sources of brGDGTs. A proxy-model comparison shows generally good agreement, and highlights the

important role of seasonality in the temperature recorded by brGDGTs. Our centennial-resolution reconstruction of TMAF indicates that early MIS-3 experienced non-freezing-season temperatures only slightly cooler than present, and that LGM TMAF was approximately 2.6 °C cooler than present. The TMAF reconstruction shows only minimal 0-1 °C temperature fluctuations across Greenland stadial-interstadial transitions, strong evidence that these millennial scale climate events primarily affect the winter season and shortened growing seasons, but peak warm-season temperatures were less affected. The

seasonal signal of TMAF provides a unique constraint on our understanding of past seasonality changes and thus may be a useful target for modeling studies of past climate. Our high-resolution, continuous temperature reconstruction is unique for the European continent and therefore will be of interest for studies of paleoecology and archaeology.



*Author contributions.* PDZ: Conceptualization, Formal analysis, Investigation, Writing - original draft, Visualization; DB:
Conceptualization, Formal analysis, Investigation; Writing – review & editing; FS: Conceptualization, Investigation, Resources, Funding acquisition, Project administration; Writing – review & editing; AA: Formal analysis, Writing – review & editing; GH: Resources, Funding acquisition, Project administration; AMG: Conceptualization, Resources, Funding acquisition, Project administration; Writing – review & editing.

*Code and data availability.* Data has been submitted to Pangaea, a DOI will be included in the published manuscript. Code can be made available on request.

*Competing interests.* The authors declare no competing interests.

*Acknowledgements.* The work was funded by the University of Mainz, Germany, the Max Planck Institute for Chemistry, Germany, and the Swiss National Science Foundation grant # P500PN_206731. We thank Klaus Schwibus, Yvonne Hamann, Brigitte Stoll, and Mareike Schmitt for technical assistance.

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
