# Peer review of "Reconstruction of warm season temperatures in central Europe during the past 60,000 years from lacustrine GDGTs"

_EGUsphere, 2023_

## Author Response (AR1)

**Response to comments from Francien Peterse**

Dear Paul Zander and co-authors,

I came across your manuscript based on your Tweet (or whatever that is called these days) and was intrigued by some of the replies that suggested a coupling between the occurrence of the IIIa'' isomer and redox changes. I just read through your work and have a few questions and comments for you to ponder on (this is not an official review, just an exchange of ideas I hope):

I thank you for voluntarily taking the time to provide useful feedback. Responses to comments are in red. Specific modifications are in blue.

Your Fig. 2 suggests that you also detected IIIa'' in soils, whereas Weber et al 2015 found that it was exclusively produced in lakes. In fact, they used the absence of IIIa'' in soils and the presence of IIIa'' in the lake sediment as their main argument for its aquatic source, later backed up by isotope analysis. Can you comment on the presence of IIIa'' in the soils studied here?

Thank you for bringing this to our attention. The level of IIIa'' in soils does not meet a robust limit of detection criteria. Our automated chromatogram evaluation picked peaks in 5 of 13 soil samples based on the retention time of the IIIa'' compound, and we applied a screening based on exceeding 3x the level of machine noise. However, due to co-elution of the IIIa and IIIa' peaks, this level can be exceeded without a robust peak. We now apply a screening based on the limit of detection as defined by Currie (1999; eq 14) based on the analysis of 17 method blanks. None of the soil samples exceed this limit, whereas all but two of our lake sediment samples do exceed this criteria. Therefore, our data strongly supports the conclusion of Weber et al. Applying this limit of detection does not alter the results or conclusions of our research.

Modifications: Inserted sentence in methods (line 144). Inserted sentence in results about which samples do or do not contain IIIa'' (line 216-218). Updated Figs 2, 3, 4, 5, and 7 to reflect the changes from applying the limit of detection.

In their follow up study (Weber et al 2018) is mentioned that IIIa'' does not occur in all lakes. Based on the isotopic composition of IIIa''-derived alkanes you could derive preferred production in eutrophic/anoxic lakes (see their Fig 4C-E), but just anoxia is not doing the trick. I can tell you firsthand that IIIa'' is not detected in Lake Chala, although this lake has anoxic bottom waters (resulting in a continuously laminated sediment record; van Bree et al., 2020, Baxter et al 2023). Maybe the fact that Chala is oligotrophic explains its absence, but I would be careful with using IIIa'' as marker for 'just' anoxic conditions.

This is a good point, and seems to require more research to fully understand the constraints on IIIa''. Our data is consistent with the interpretation of a connection with trophic levels as we see higher IIIa'' during times of higher productivity (indicated by org C/chloropigment data). It seems that the presence of IIIa'' is a useful indication that anoxic conditions were present, but your Challa data clearly show that the absence of IIIa'' is not necessarily an indication of the absence of anoxic conditions. We will clarify this interpretation in the text.

Modification: Added sentence about possibility of nutrient levels influencing IIIa'' abundance (line 241-243).

Based on all the lake work that has recently appeared in the literature I have started to have serious doubts about the importance of soil-derived GDGTs in lake sediments, especially for lakes that are not connected to any river system. I feel that the vast majority, if not all brGDGTs in lake sediments are produced in situ, and that their production in the water column is strongly impacted by redox conditions, as you already mention in your manuscript, and has also clearly been shown by e.g., Weber et al., 2018, van Bree et al., 2020, Wu et al 2021, and Baxter et al., 2023. Most importantly, these studies have also shown that the producers of the different brGDGT isomers may occupy different niches in the water column. This can make brGDGT production very sensitive to changes in e.g., lake depth (be it climate driven, or by lake basin development or infilling of the lake over time) or mixing regime and affect the reliability of the currently available transfer functions. I know this mostly from working on Lake Chala, where we did very extensive lake monitoring to find this out (see Baxter et al., 2023), and I can imagine that this is not possible for every lake, but we should try to use all available information from these studies in our interpretation of novel lake records. Enfin, the point I wanted to make was that changes in IIIa/IIa may not necessarily reflect soil vs aquatic brGDGT sources, but instead the expansion and shrinking of the niches of their producers within the lake, related to redox. The different zones in the water column occupied by the producers of IIa and IIIa is actually clearly visible in your Fig. 4e.

Following this line or reasoning and letting go of IIIa/IIa as indicator for soil input, this also allows you to interpret the high BIT index values during the Holocene differently, especially since the high BIT index values are not supported by more clastic material (Fig. 7I), but rather with increased productivity, sediment anoxia, and methanogens (Fig. 7GHD). This would fit with a shallower oxycline in the lake, which increases the zone for brGDGT production (in the anoxic zone of the water column; see Weber et al., 2018, van Bree et al., 2020, Wu et al., 2021), and decreases the zone of crenarchaeol production, which, under ideal conditions peaks in the suboxic zone, as is also the case in the modern lake, shown in your Fig. 4C. As Thaumarchaeota exhibit photoinhibition, they cannot thrive in the very upper water layer, and are thus outcompeted during periods with less mixing, more stratification, etc, increasing the BIT index (see Baxter et al., 2021 and 2023 for detailed explanation and illustrating schematics). That this scenario may hold true is also reflected by the absolute abundances of the GDGTs, which show an increase in brGDGTs and isoGDGTs that are produced in the anoxic part of the lake, whereas crenarchaeol remains relatively stable/low during the transition into the Holocene.

I haven't looked in much detail at the part of the record spanning MIS3, so I am not sure what exactly is going on there, and I can only encourage you to keep the water column structure, mixing regime, and lake basin development in the back of your mind when interpreting the data! Looking forward to hearing your thoughts about all of this.

I generally agree with your points about IIIa/IIa and BIT reflecting changes in the environmental niche (mainly redox) of the producers within the lake. We emphasized these indices in the paper as indicators of soil input because they have been the most often cited in the literature, and I think they can be useful in that regard in some settings. But I agree with you that these indices cannot be interpreted as being driven by soil input alone.

Modification: Added text to section 3.2 about other factors influencing IIa/IIa (Lines 363-365).

All the best,

Francien Peterse

PS. Quick last note for the introduction: the presence of brGDGT Ia in culture was already shown by Sinninghe Damsté et al in 2011. Also, I miss a reference to Weijers et al., 2007 as the paper in which the temperature dependency of brGDGTs was first demonstrated and the MBT was defined.

Modification: references added to introduction

References:

Baxter et al 2021 QSR

Baxter et al 2023 Nature (check methods section on calibrations and extended data Fig. 2)

van Bree et al 2020 BGS

Sinninghe Damsté et al., 2011 AEM

Weber et al 2015 GCA

Weber et al 2018 PNAS

Weijers et al., 2007 GCA

Wu et al 2021 Chemical Geology

Thank you for the productive exchange and useful reference suggestions, these will also be incorporated in a revised version of the manuscript.

On behalf of all authors,

Paul Zander

References

Currie, L. A.: Nomenclature in Evaluation of Analytical Methods including Detection and Quantification Capabilities, Analytica Chimica Acta, 391, 105–126, https://doi.org/10.1016/S0003-2670(99)00104-X, 1999.

**Response to review from Maria Fernanda Sanchez Goñi**

This manuscript provides a continuous record from 60 ka to the present of brGDGT temperature reconstruction for the Eifel region (Central Europe). This record is relevant and deserves to be published in Climate of the Past. So far, there is a lack of regional quantitative climate reconstructions for the last glacial period in Central Europe and very few for other European regions. However, before accepting this manuscript for publication, the authors should address some chronological issues and discuss the ELSA record in more detail, taking into account the other quantitative climate reconstructions for Europe. I am not an expert in geochemistry and my review is mainly focused on section 3 of the manuscript "Results and discussion".

We appreciate the constructive comments, and our responses and completed edits are in red below.

Lines 35-40 and lines 450-453: In the Introduction and in Section 3.4, it would be relevant to cite the papers that, based on pollen analysis of deep-sea sediment sequences, have detected for the first time, without chronological ambiguity, the European millennial scale climatic variability in response to the D-O warming and cooling events, including the Heinrich events (e.g. Sanchez Goñi et al. 2000, 2008). The only works cited in the manuscript are the terrestrial pollen records and speleothem sequences, which, even if relatively well dated, cannot be directly compared with the North Atlantic millennial-scale climate variability.

I agree the mentioned papers are important to cite as they established a definitive link between European terrestrial environment changes and N. Atlantic millennial-scale events.

Modification: Citations added in several places in the Introduction and discussion (Line 40, 470, 486)

Figure 7: How is it possible to have 50%-100% of tree pollen in the 60-30 ka interval? These values are found during interglacial periods, as shown during the Holocene of the same sequence.

The pollen data are published in Sirocko et al. (2022) and the chronology is well described in Sirocko et al. (2021). At this site, evidence suggests that MIS 3 climate was relatively mild, allowing for the growth of temperate forests from 58-48 kyr BP. After 41 kyr BP pollen suggest a steppe or woodland-steppe environment. The temperate forest in early MIS 3 fits with our GDGT data which suggest that early MIS 3 growing season temperatures were not significantly different from present day. It is always possible that tree pollen data are somewhat inflated due to long distance transport of pollen.

Lines 394-401 in section 3.3 and lines 453-471 in section 3.4: I do not know to what extent the temperature of months above freezing (TMAF) in the Eifel region can be compared with the sea surface temperatures (SST) of the Iberian margin but if the authors do so, I think it is more appropriate to discuss the TMAF in the Eifel region in the light of the surface air temperatures estimated from the pollen records of the Iberian margin, the Alboran Sea (Sanchez Goñi et al., 2002) and the Gulf of Lyon (Sanchez Goñi et al., 2021), which cover the period 50-27 ka, i.e. HS 5 to HS 3 and the associated D-O cycles. Although the latter correspond to winter temperatures, the authors should compare their estimates of summer temperatures with them, especially in the light of the question of seasonality they discuss in Section 3.4, which is the main contribution of this work. For HS 5, pollen-based winter surface air temperatures show pronounced decreases, averaging 5-10°C relative to present, in southern Iberia and south-eastern France, consistent with the severe SST low in the eastern North Atlantic. HS 3 is also marked by cooling in southern Iberia, although less so than HS 5 and HS 4. Furthermore, why are HS 5 and HS 3 marked by warm summers in the Eifel region and HS 4 by cold summers? Are these seasonal

changes linked, for example, to orbital parameters?Figure 10 should be implemented with the pollen-based surface air temperatures from western Europe.

This is a good point. We will incorporate discussion of the pollen-based temperature estimates.

Modification: We have modified Figure 10 to include the mentioned pollen data. We have added text in section 3.4 to compare our results with the pollen temperature estimates (Line 484-487).

Lines 409-422: The authors discuss the mean temperature of the 29-24 kyr b2k interval in the Eifel and compare it with other reconstructions for the LGM without defining their respective chronologies. This interval extends from 29 to 19 ka according to Hughes et al. (2022) but from 23 to 19 ka according to Mix et al. (2001, EPILOG). The problem is that the LGM according to Hughes et al. encompasses three D-O warming events (D-O 4, D-O 3 and D-O 2), HS 2 and the LGM defined by EPILOG. Do they compare the same period in the different reconstructions (Paris Basin and Auel Maar, "...considering the significant age uncertainty in the Auel Maar data", line 372, for example), the assimilation of global palaeoclimatic data and the output of the HadCM3 model? In terms of vegetation in north-western and southern Iberia, the HS and LGM are markedly different (Naughton et al., 2007; Fletcher et al., 2008; Turon et al., 2003), with more Ericaceae and less Amaranthaceae-Chenopodiaceae during the LGM compared with the HS that flank it, indicating drier and colder conditions during the HS, associated with greater seasonality.

In lines 409-422 and Figure 11 we compare maximum LGM cooling relative to present for several datasets. We aim to use the same time window for comparison based on the coldest period in the GDGT data (29-24 kyr BP). This interval was used to estimate the LGM temperature from the HADCM3 simulation. The Paris Basin noble gas data (Bekaert et al., 2023) shows a temperature minimum around 26 kyr BP (low-resolution data). The Tierney et al. (2020) data assimilation uses 23-19 kyr BP, so this dataset comes from a slightly different time period. However, it should be noted that the emphasis here is on comparing a sustained period of minimum temperatures, and the datasets we compare with do not resolve millennial-scale variability. We will edit the text to note the ages of the temperature estimates from (Bekaert et al., 2023; Tierney et al., 2020).

Modification: Dates have been added to the text for the LGM cooling estimates from other sources (Line 431-432).

Lines 468-471 : It is interesting to add that the 17°C of warming of the coldest month between stadial and interstadial yielded by the LOVECLIM model simulations (Van Vermeerck et al., 2011) is higher compared to the pollen-based estimations of 10°C on average for SW and SE Iberia and 5°C for SE France.

Agreed that this particular climate model simulation yields a rather large stadial-interstadial amplitude.

The last part of section 3.4, devoted to the examination of changes in seasonality during the last glacial period, should be improved and strengthened by taking into account the pollen-based reconstructions of winter temperatures available for Western Europe.

Agreed, we will implement this suggestion and include mention of winter temperature reconstructions based on pollen, which strengthen the argument that stadial-interstadial changes were dominated by winter temperature changes.

Modification: We have added text (lines 484-487 and 516-519) to incorporate the pollen-based reconstructions into the discussion.

Minor comments

Line 212 : Specify in the text the meaning of the abbreviation SM = Schalkenmehrener Maar

Line 214 : Replace Schalkenmehrener Maar with « SM »

Lines 245-247 : Rephrase. A dot is lacking after Figure 5b.

Line 391 : Replace « prescence » with « presence ».

Line 445 : the subject of the sentece is lacking.

Modification: All minor comments have been corrected.

References

Fletcher, W.J. & Sanchez Goñi, M.F. (2008) Orbital- and sub-orbital-scale climate impacts on vegetation of the western Mediterranean basin over the last 48,000 yr, Quaternary Research, 70, 451-464.

Hughes et al. (2022) The European glacial landscapes from the Last Glacial Maximum – synthesis. In : European Glacial Landscapes Maximum Extent of Glaciations, Elsevier, Pages 507-516.

Mix, A., Bard, E., Schneider, R., (2001). Environmental processes of the ice age: land, oceans, glaciers (EPILOG). Quat. Sci. Rev. 20, 627–657.

Naughton, F., et al. (2007). Present-day and past (last 25 000 years) marine pollen signal off western Iberia. Marine Micropaleontology 62: 91-114.

Sánchez Goñi, M.F., et al. (2000). European climatic response to millenial-scale changes in the atmosphere-ocean system during the Last Glacial period. Quaternary Research 54: 394-403.

Sanchez Goni M.F., et al. (2008) Contrasting impacts of Dansgaard-Oeschger events over a western European latitudinal transect modulated by orbital parameters, Quaternary Science Reviews, 27, 1136-1151.

Sánchez Goñi, M.F., et al. (2021). Muted cooling and drying of NW Mediterranean in response to the strongest last glacial North American ice surges. Bulletin of the Geological Society of America 133, 451-460.

Turon, J.-L., Lézine, A.-M., Denèfle, M., (2003). Land–sea correlations for the last glaciation inferred from a pollen and dinocyst record from the Portuguese margin. Quat. Res. 59, 88–96.

We thank the reviewer for the suggestions, which will be implemented in a revised manuscript.

On behalf of all authors, Paul Zander

References

Bekaert, D. V., Blard, P.-H., Raoult, Y., Pik, R., Kipfer, R., Seltzer, A. M., Legrain, E., and Marty, B.: Last glacial maximum cooling of 9 °C in continental Europe from a 40 kyr-long noble gas paleothermometry record, Quaternary Science Reviews, 310, 108123, https://doi.org/10.1016/j.quascirev.2023.108123, 2023.

Sirocko, F., Martínez-García, A., Mudelsee, M., Albert, J., Britzius, S., Christl, M., Diehl, D., Diensberg, B., Friedrich, R., Fuhrmann, F., Muscheler, R., Hamann, Y., Schneider, R., Schwibus, K., and Haug, G. H.: Muted multidecadal climate variability in central Europe during cold stadial periods, Nature Geoscience, 14, 651–658, https://doi.org/10.1038/s41561-021-00786-1, 2021.

Sirocko, F., Albert, J., Britzius, S., Dreher, F., Martínez-García, A., Dosseto, A., Burger, J., Terberger, T., and Haug, G.: Thresholds for the presence of glacial megafauna in central Europe during the last 60,000 years, Sci Rep, 12, 20055, https://doi.org/10.1038/s41598-022-22464-x, 2022.

Tierney, J. E., Zhu, J., King, J., Malevich, S. B., Hakim, G. J., and Poulsen, C. J.: Glacial cooling and climate sensitivity revisited, Nature, 584, 569–573, https://doi.org/10.1038/s41586-020-2617-x, 2020.

**Response to review from Anonymous Reviewer 2**

The manuscript by Zander et al. presents a 60 kyr long record of brGDGTs composed by several lake sediment records from the Eifel volcanic field, Germany. By using an array of modern soil and top core sediments from the area, the authors show a clear and unique lacustrine signal in the lake sediments. Additionally this modern dataset performs relatively well when recent temperature calibrations are applied to it, however the authors observe that the performance of this calibrations is enhanced when a correction to the calibration is created based on the abundance of the isomer IIIa''. When the Raberg et al., 2021 calibration, adjusted with the IIIa'' correction, is applied to the 60 kyr record; the authors observe a generally good agreement between this record and previously published records for this period. Nevertheless, warmer than expected temperatures during Dansgaard-Oeschger events are interpreted by the authors as the proxy recording generally stable non freezing season temperatures, while the DO influence may have primarily been reflected in the winter season in this area.

I think this is a very well written manuscript with a clear structure and well presented data and results. I do not have any major comment and would be happy to recommend it for publication once the following minor observations are addressed.

Thank you for your positive review, responses are in red below

Figure 3 A and B: I imagine the authors might have gone for open shapes in these panels to avoid cluttering the figure. However I find it slightly difficult to distinguish the colors when only the outlines are presented. Maybe using some transparency could help avoid cluttering in this figure and would make it easier to read.

Modification: We have revised the figure to use filled shapes to improve the clarity of the figure.

Lines 287-288: Could you mention in-text some of the most relevant correlation values? I think it would enhance the reading of the text without having to look for Table 2.

Yes, thank you for the helpful comment. We will insert the most important correlation results into the text.

Modification: Correlation coefficients are now inserted (lines 296-301).

Figure 8: This figure is a little hard to read since all the values from the different bins are presented together and it makes it hard to distinguish which bin the points belong to. Maybe something like removing the horizontal lines could improve this, or adding some space between the bins, would be my suggestion.
We will revise the figure to include more space between bins to improve readability.

Modification: We have attempted to improve the readability by making the points more distinct and relabeling the y-axis.

Line 445: Sentence is missing the first part.

Lines 459-460: This sentence is a little hard to read, please rephrase it.

Modification: The text has been edited to fix the above comments.

---

## Author Response (AR2)

**Reply to Jan 9 report from Maria Fernanda Sanchez Goñi**

In the revised manuscript the authors have responded well to my comments and it is ready for publication after correcting the following minor problems:

a) In Figure 10, the pollen-based MTCO reconstructions from core MD95-2042 should be plotted against the updated age model that is in line with the chronology of Heinrich stadials published in Sanchez Goñi and Harrison (2010, QSR).

The figure has been updated with the latest chronology. I thank Dr. Sanchez Goñi for sending the updated data file.

b) In line 506, the DJF cooling is stronger, between 1.5°C and 4°C, compared to the interval given by the authors, 1.5-2.5°C.

This has been corrected in the text.

c) In the legend of Figure 6, the pink bands indicate the Heinrich stadials (HS) and not Heinrich events (HE). The Heinrich stadial is the climate interval associated with the Heinrich events (massive iceberg discharges from the North American ice-sheets during the last glacial period). The authors should modify the legend and the figure accordingly. The same applies for the text, for example lines 429, 506, 524, 528 and 541, where the authors should replace Heinrich event with Heinrich stadial.

We have replaced Heinrich event with Heinrich stadial throughout, unless referring specifically to iceberg discharge events.

d) In the figures, the width of the bands indicating the time intervals of the Heinrich stadials should take into account the chronology of these intervals, broadly accepted, given by Sanchez Goñi and Harrison (2010, QSR)

In general, we agree; however, the age of H3 in Sanchez Goñi and Harrison (2010) seems to be not widely accepted, as it does not fit with Iberian margin SSTs or Greenland ice core records of temperature and dust. Therefore, we use the age 30.6–28.9 b2k, which corresponds to Greenland stadial 5 (Rasmussen et al., 2014). We have updated the Heinrich bands (except H3) to match the ages of Sanchez Goñi and Harrison (2010) in all figures.

**Reply to 02 Feb report by Anonymous referee #3**

I am providing a review of the author replies to previous reviewers' comments and the adjustments made in the manuscript. I think that the authors have done a good job responding to the main concerns and my additional points are only minor. Notably, temperature reconstruction based on brGDGTs in lacustrine sediments is notoriously hard, so I appreciate the efforts of the authors here. Although they have done an above average job in assessing the brGDGT signals in their record, I feel like there could have been even more. Regardless, given the overall positive feedback in the first round of comments, I'll leave it to future studies with further insights into lacustrine brGDGT dynamics to possibly extract this information from the data on Pangaea.

L18: We find a bias… -> specify: bias in what?

Text modified to read "We find a negative bias in brGDGT-based temperature estimates"

L42§ion 3.4: Millennial scale climate events are also detected in Lake Van (there are multiple papers from this lake, but Stockhecke et al. (2021, https://doi.org/10.1016/j.palaeo.2021.110535) specifically is on biomarkers and D-O events, but just predates the global lake calibration – still, D-O variation is presumably >>1C!)

Added citation in the Introduction and section 3.4

L142: were one (as is mentioned in the manuscript) or two (as is proposed by Hopmans et al., 2016) silica columns used for GDGT separation by the LC?

Two columns, thank you for pointing out this error, which has been fixed.

Calibration selection: I never really understand the urge to assess all possible calibrations and then continue with the one that looks most like the expected (or desired?) record. How does matching expectations validate the reliability of the record? Especially when most of the tested calibrations come down to (some form of) the MBT…

There is always information in downcore changes in brGDGT (or any biomarker) distributions. So, if one calibration does "not work" it is always worth to investigate which compound is driving this and then to assess why this compound could be behaving the way it is. Notably, the PCAs in Fig. 3 suggest that temperature is likely not recorded well by the MBT'5me, as the compounds that are in the numerator (tetramethylated 5-me brGDGTs) and the denominator (penta- and hexamethylated 5-me brGDGTs) of this ratio do not clearly plot in opposite parts of any of the PCs (specifically, IIa plots with all tetramethylated brGDGTs, which would make the MBT'5me mostly dependent on IIIa, IIIb, and IIIc).

Anyhow, I guess that this manuscript has passed the stage of this kind of comments, so keep things as is. Also, during the review process of this manuscript two new lacustrine brGDGT studies have appeared: one specifically for European lakes (Bauersachs et al., 2024, https://doi.org/10.1016/j.scitotenv.2023.167724) and a global one (O'Beirne et al., 2023, https://doi.org/10.1016/j.gca.2023.08.019) that both address some of the sensitivity issues that are also discussed in the current manuscript. In particular the European calibration study addresses the potential role and presence of IIIa'' in lake sediments.

I note that while many calibrations were tested on modern samples, only two published calibrations were tested on the downcore sequence (Martínez-Sosa et al., 2021; Raberg et al., 2021), and these two have quite different formulas, and therefore temperature reconstructions can differ substantially.

Its unclear to me if the reviewer would like us to test the more recently published calibrations. This would represent a major revision at this late stage in the review process, and given that new calibrations are published frequently, I think it is reasonable to only consider those that were published before the paper was submitted. As a note, the new European calibration is based on the Ia brGDGT and will likely yield results very similar to the temperatures we use from the corrected Raberg21 model based on the strong correlation between Ia and our reconstructed TMAF, as shown in the PCA (Fig 3).

L370: Add reference after the statement that BIT is more likely controlled by the niche for crenarchaeol producing Thaumarchaeota (which are currently named Nitrososphaerota btw)

Reference and additional explanation added. And Thaumarchaeota has been updated to Nitrososphaerota.

L378: I agree with the interpretation that IIIa/IIa is more likely to reflect contributions from different niches within the water column than soil vs aquatic producers. Please also adjust this in the panels and captions of the figures (for those people that 'read' a paper by only looking at the graphs). Also, later in the discussion (e.g., section 3.3) the interpretation of IIIa/IIa as a soil indicator is still maintained. So, which interpretation is considered valid in the end?

Difficult to say which is valid in the end! Maybe a bit of both. Our data clearly show soils do have lower IIIa/IIa. The increase in the ratio with depth in our lake sediment transect could be driven by decreasing soil input with greater distance from shore, or by changes in aquatic communities with depth (likely both?). Nonetheless, the data from global compilations of lacustrine brGDGTs show that the thresholds from the marine environment from (Xiao et al., 2016) might not be applicable to lakes. Therefore, we have de-emphasized the interpretation of IIIa/IIa as a soil indicator by removing the thresholds in Fig. 4, and changing the labeling of the index in Fig. 7.

General comments:

Replace all apostrophes with the prime symbol throughout the manuscript.

Fixed

Fig. 4: connect datapoints with straight lines as to not infer trends.

Due to the fact that there are multiple data points at the same depth, connecting points with straight lines requires some arbitrary choices of data point ordering and can lead to false impressions of trends. We have modified the plot by using generalized additive models as a smoothing algorithm, replacing the previously used LOESS method. The GAM algorithim is less flexible and thus reduces the curviness of the lines. We have also reduced the line thickness to de-emphasize the lines relative to the data points.

Check the order of the figures to match with their occurrence in the text. For sure Fig. 3 is out of order.

Figure 3 is first mentioned in the first sentence of the results (line 215), thus it is in the correct order.